# 1.25(OH)$_2$D$_3$ decreases PCNA and mTOR expression and alleviates renal injury in Thy-1 nephritis rat model

Jian-feng Li[1‡], Lei Jin[2‡], Huan Ma[3], Jie Suo[4], Rui Yang[5], Xiao-ping Yang🔘[5]*

1 Department of Nephrology, The First People's Hospital of Nanyang City, Nanyang, Henan Province, China,
2 Department of Rheumatology, Immunology & Allergy, Shanghai General Hospital, Shanghai Jiao Tong University School of Medicine, Shanghai, China, 3 Department of Gastrointestinal Surgery, The First People's Hospital of Nanyang City, Nanyang, Henan Province, China, 4 Department of Nephrology, The First People's Hospital of Aksu City, Aksu, Xin jiang Province, China, 5 Department of Nephrology, The First Affiliated Hospital of Shi he zi University, Shihezi City, Xin jiang Province, China

‡ These authors contributed equally to this work and share first authorship
* xjshzdxyxp@163.com

## Abstract

### Objective

This study investigated the role and mechanisms of 1.25(OH)$_2$D$_3$ in proliferative glomerulonephritis and its effect on the regulation of mesangial cells.

### Methods

Sixty male SD rats were randomly divided into four groups: control (CG), nephritis (NG), nephritis + 1.25(OH)$_2$D$_3$(NVG), and nephritis + 1.25(OH)$_2$D$_3$+ rapamycin (NVRG) (n = 15 per group). Three rats from each group were sacrificed on days 1, 3, 7, 14, and 21 after intervention. Urine samples were collected over 24 hours on day 0 to measure urinary protein excretion. Renal tissue samples were stained with HE and PAS to evaluate the extent of renal injury, while immunohistochemistry was employed to quantify PCNA and mTOR expression in the renal tissues.

### Results

Compared to the NG, mesangial cell proliferation in the renal tissues was significantly reduced in the NVG and NVRG at all time points (all p<0.05). PCNA expressionwas significantly higher in the NG compared to the CG (p < 0.05) and significantly lower in the NVG and NVRG (p < 0.05). mTOR expression was also significantly increased in the NG compared to the CG, with a significant reduction observed in the NVG and NVRG compared to the NG.

### Conclusion

Our findings demonstrate that 1.25(OH)$_2$D$_3$significantly inhibits the proliferation of glomerular mesangial cells in rats. Additionally, mTOR protein is involved in the regulation of

**Data Availability Statement:** All relevant data are within the manuscript.

**Funding:** This work was supported by the National Natural Resources Foundation of China (81160090) and Research Project Supported by Xinjiang Production and Construction Corps of China (2009GG59) The funders had no role in study design, data collection and analysis, decision to publish, or preparation of the manuscript.

**Competing interests:** The authors have declared that no competing interests exist.

**Abbreviations:** CG, Control Group; NG, Nephritis Group; mTOR, mammalian target of rapamycin; ESRD, end-stage renal disease; PI3K, phosphoinositide 3-kinase; HE, Hematoxylin and eosin; MPGN, mesangial proliferative glomerulonephritis; PCNA, proliferating cell nuclear antigen.

glomerular mesangial cells by $1.25(OH)_2D_3$. These results further elucidate the molecular mechanism by which $1.25(OH)_2D_3$ alleviates renal injury in glomerulonephritis.

## Introduction

Glomerulonephritis is a common cause of end-stage renal disease (ESRD) [1]. Data from the National Dialysis Patient Registry of the China Renal Data System indicate that glomerulonephritis is the leading cause of ESRD in China [2, 3]. The key pathological features of glomerulonephritis include mesangial cell proliferation and abnormal extracellular matrix deposition. Early treatment of nephrotic syndrome with steroids can lead to better long-term remission [4]. However, in patients with membranous glomerulopathy or proliferative nephritis, steroids alone have shown only modest improvement in renal parameters [5]. Therefore, it is critically important to explore novel therapeutic agents for treating mesangioproliferative glomerulonephritis and preventing ESRD.

1.25-Dihydroxyvitamin D3 ($1.25(OH)_2D_3$), the active form of vitamin D3, has been reported to possess anti-inflammatory, anti-tumor, and cardiovascular-protective properties [6–8]. Proper treatment can induce apoptosis in proliferated mesangial cells, facilitating the recovery of glomerular structure, as demonstrated in the Thy-1 nephritis model [9]. Inhibiting mesangial cell proliferation can delay disease progression and improve prognosis. Previous studies have shown that $1.25(OH)_2D_3$ can significantly inhibit the proliferation of mesangial cells cultured in vitro and promote their differentiation and apoptosis [10, 11]. However, it remains unclear whether $1.25(OH)_2D_3$ can regulate the pathology and progression of glomerulonephritis through mTOR.

mTOR has been reported to be involved in mesangial cell proliferation and oxidative damage in the development of chronic kidney diseases, such as diabetic nephropathy and IgA nephropathy [12, 13]. Activation of the phosphoinositide 3-kinase (PI3K)/protein kinase (Akt)/mammalian target of rapamycin (mTOR) signaling pathway can influence cell proliferation, hypertrophy, and apoptosis [14]. However, the role of the mTOR signaling pathway in the regulation of mesangial cells by $1.25(OH)_2D_3$ has been rarely explored.

In this study, we established Thy-1 rat nephritis models and administered $1.25(OH)_2D_3$ and rapamycin to the experimental groups to observe changes in PCNA and mTOR expression in rat renal tissues at different time points. We also conducted a preliminary investigation into the target and mechanism of $1.25(OH)_2D_3$ in regulating mesangial cells, aiming to provide a reliable theoretical basis for further research in this field.

## Methods

### Animals

Sixty male SD rats (6 weeks old, weighing 180–200 g) were obtained from the Xinjiang Center for Local Epidemic Disease Control. The rats were housed in a controlled environment with a temperature of $22 \pm 1$ ºC, relative humidity of $50 \pm 1\%$, and a 12/12-hour light/dark cycle, with noise levels kept below 85 dB. They had free access to food and water. All animal procedures, including euthanasia, were approved by the Shihezi University Institutional Animal Care and Use Committee and conducted in accordance with the regulations and animal welfare guidelines of Shihezi University (SCXK [Xin] 2013–178). The animals were randomly assigned to four groups (15 rats per group): control group (CG), nephritis group (NG), nephritis + 1.25 $(OH)_2D_3$ group (NVG), and nephritis + $1.25(OH)_2D_3$ + rapamycin group (NVRG).

## Thy-1 nephritis induction and treatment

Thy-1 nephritis models were established as previously described [15]. Briefly, rats in the NG, NVG, and NVRG groups received a single tail vein injection of 25 μL/100 g of Thy-1 monoclonal antibodies (Ontario, Canada, catalog No.: CL005A), while rats in the CG received an equivalent dose of saline. After injection, urine was collected from all groups over 24 hours using metabolic cages, and proteinuria in fresh urine was assessed using the standard Coomassie blue staining assay. Following successful model induction, rats in the NVG group received a daily gastric perfusion of 0.5 μg of $1.25(OH)_2D_3$ (Shanghai Roche Pharmaceuticals Ltd.) dissolved in 1 mL of peanut oil, while rats in the NVRG group received a daily gastric perfusion of 0.5 μg of $1.25(OH)_2D_3$ and 1 mg of rapamycin (Yixinke, Research Institute of North China Pharmaceutical Group) dissolved in 1 mL of peanut oil. Rats in the CG and NG groups were given a daily gastric perfusion of 1 mL of peanut oil. The Thy-1 nephritis model exhibits self-healing properties, so observations were made on days 1, 3, 7, 14, and 21. On these days, 3 rats from each group were randomly sacrificed using 2% pentobarbital sodium (30 mg/kg, intraperitoneally), and kidney tissues were collected.

## Hematoxylin and eosin (HE) staining and Periodic Acid-Schiff (PAS) staining

For HE staining, renal tissues were fixed in 10% formaldehyde, embedded in paraffin, and sectioned into 3-μm-thick slices. The sections were baked at 60°C for 2 hours, then placed in xylene and ethanol for dewaxing and hydration. They were stained with hematoxylin for 5 minutes and eosin for 3 minutes. For PAS staining, sections were treated with periodic acid, washed, incubated with Schiff's reagent, and then stained with hematoxylin. After dehydration with ethanol and xylene, the sections were sealed with neutral gum.

Pathological damage was graded according to the criteria for human mesangial proliferative glomerulonephritis (MPGN) as follows: grade 0, normal; grade I, mild mesangial proliferation; grade II, open capillary loops with mesangial width not exceeding the capillary diameter, showing segmental distribution, and moderate mesangial proliferation; grade III, compressed capillary loops with mesangial width exceeding the capillary diameter, showing diffuse distribution, severe proliferation, and severely compressed capillary loops, along with the presence of nodules and clumpy solid areas.

## Immunohistochemistry

Immunohistochemistry was performed using a universal two-step PV-9000 test kit (ZSGB-BIO, China) following the manufacturer's instructions. PCNA antibodies (#2586, 1:50) and mTOR antibodies (#2983, 1:2000) were obtained from Cell Signaling Technology. Five fields of view at 400X magnification were randomly selected from each sample to assess the percentage of positive cells and staining intensity. The percentage of positive cells was scored as follows: 0 point for no staining, 1 point for <10% staining, 2 points for 10%-50% staining, and 3 points for >50% staining. Staining intensity was scored as follows: 0 point for no staining, 1 point for light brownish-yellow staining, 2 points for brownish-yellow staining, and 3 points for dark brown staining. The final score was calculated by multiplying the percentage of positive cells score by the staining intensity score: negative (-) with a final score of 0; weak positive (+) with a final score of 2–3; moderate positive (++) with a final score of 4–5; and strong positive (+++) with a final score of 7–9.

## Statistical analysis

All data were presented as mean ± standard deviation and analyzed using SPSS software (version 13.0; SPSS, Inc.). A paired t-test was used for paired samples. A nonparametric test was employed to compare the expression levels of PCNA and mTOR proteins at different time points and among different groups. A P-value of <0.05 was considered statistically significant.

## Results

### 1.25(OH)$_2$D$_3$ reduces proteinuria levels of Thy-1 nephritis rats

To assess whether 1.25(OH)$_2$D$_3$ can alleviate nephritis, proteinuria levels were measured in the CG, NG, NVG, and NVRG groups on days 0, 1, 3, 7, 14, and 21. The amount of water intake and urine volume across the four groups, as measured in metabolic cages, showed no statistically significant differences (p >0.05). Changes in urine protein levels are summarized in Table 1. On day 1, the NG, NVG, and NVRG groups exhibited significantly higher proteinuria levels compared to the CG, confirming the successful establishment of the Thy-1 nephritis rat model. By days 3, 7, and 14, proteinuria levels were significantly reduced in the NVG and NVRG groups relative to the NG, indicating that 1.25(OH)$_2$D$_3$, both alone and in combination with rapamycin, effectively ameliorated Thy-1-induced nephritis (Fig 1 and Table 1).

### 1.25(OH)$_2$D$_3$ mitigates renal injury in the Thy-1 induced nephritis rat model

The effect of 1.25(OH)$_2$D$_3$ on renal pathology was further evaluated using HE(Fig 2) and PAS (Fig 3) staining. The renal tissues of the CG group appeared normal, with no evidence of mesangial cell proliferation or extracellular matrix deposition, resulting in a grade 0 for pathological damage. In contrast, the NG group exhibited significant mesangial cell proliferation, severe compression of capillary loops, and the presence of nodules and clumpy solid areas at all time points, with the most severe pathological injury observed on day 7, followed by some alleviation by day 21. Additionally, 4 rats(from days3,7,14) in the NG group displayed glomerular lobulation, sclerosis, and fibrosis. Compared to the NG group, the NVG and NVRG groups showed significantly reduced mesangial cell proliferation at all time points (all p < 0.05), particularly on days 7 and 14. In these groups, capillary loops were only mildly compressed, with the mesangial width not exceeding the capillary diameter, and showed a segmental distribution (Table 2). These findings suggested that 1.25(OH)$_2$D$_3$ reduced renal injury, including mesangial cell proliferation.

**Table 1. Effect of 1.25(OH)$_2$D$_3$ treatment on proteinuria at different time points after the induction of glomerulonephritis.**

|      | day 0       | Day 1         | Day 3        | Day 7        | Day 14        | Day 21      |
|------|-------------|---------------|--------------|--------------|---------------|-------------|
| CG   | 1.517±0.712 | 1.119±0.434   | 1.528±0.107  | 2.748±0.569  | 1.769±0.174   | 2.54±0.439  |
| NG   | 1.139±0.286 | 4.896±0.539*  | 10.98±0.156  | 8.04±1.699   | 4.45±0.354    | 2.482±0.594 |
| NVG  | 0.786±0.181 | 3.015±0.397*  | 3.865±0.74#  | 5.53±1.057#  | 2.11±0.128#   | 2.241±0.688 |
| NVRG | 1.056±0.674 | 3.283±0.205*  | 3.157±0.205# | 4.511±0.781# | 1.808±0.593#  | 1.463±0.879 |

Note: *p<0.05, compared to CG, indicates significant higher proteinuria on Day 1 after antibody injection. *p<0.05, compared to CG, indicates significantly decreased mean urinary protein level of NVG on Day 3, Day 7, and Day 14. Unit: mg/d

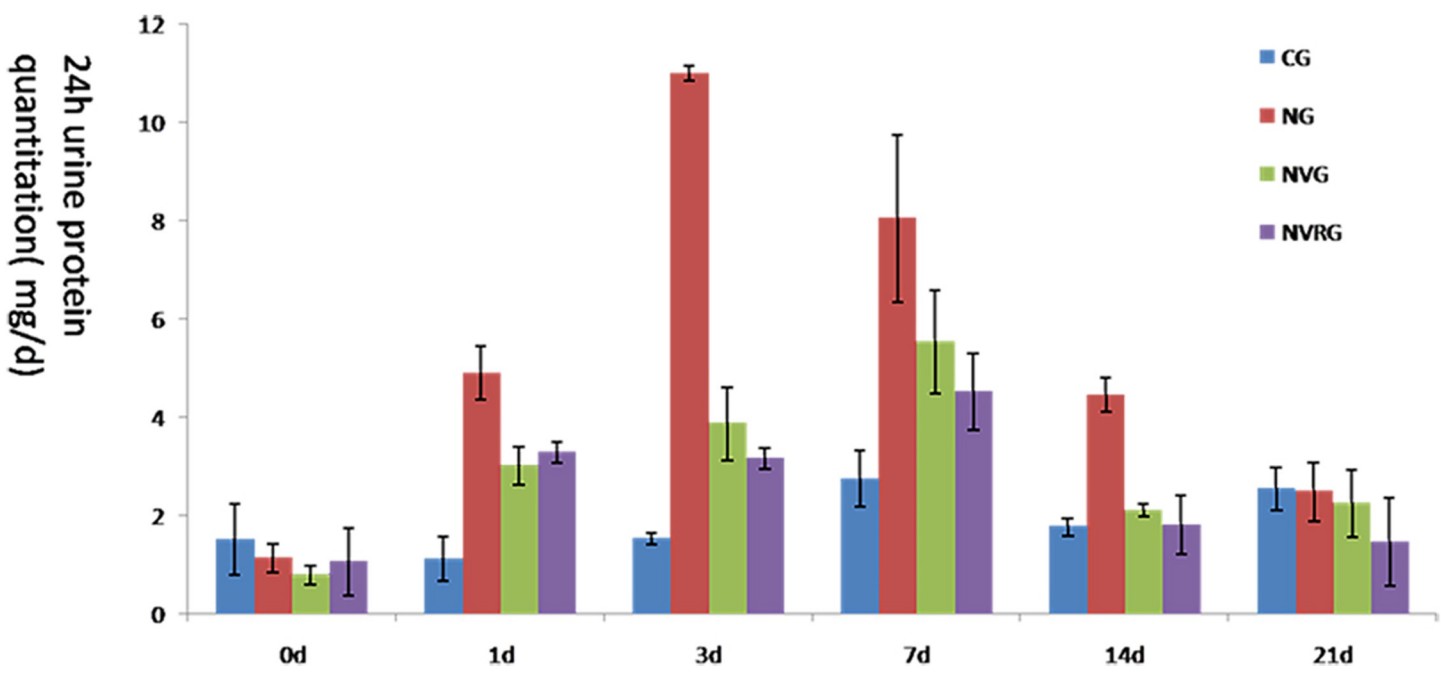

**Fig 1. Measurement of proteinuria levels in CG, NG, NVG, and NVRG.**

## PCNA and mTOR expressions measured by immunohistochemistry

The results from Tables 1 and 2 suggest that1.25(OH)$_2$D$_3$effectively suppresses proteinuria and reduce mesangial cell proliferation and abnormal extracellular matrix deposition, demonstrating its protective effects on renal function. To investigate the underlying mechanisms, we

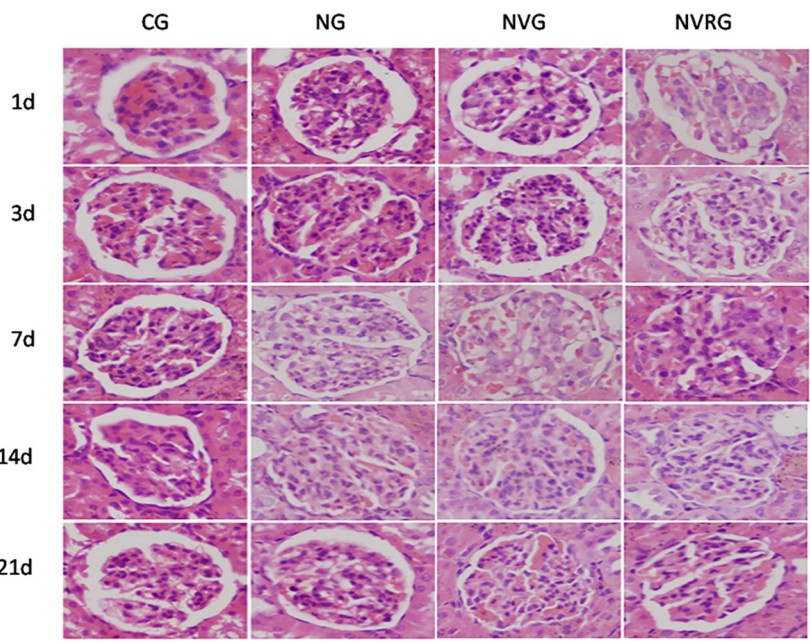

**Fig 2. HE staining images of renal tissues (×400).**

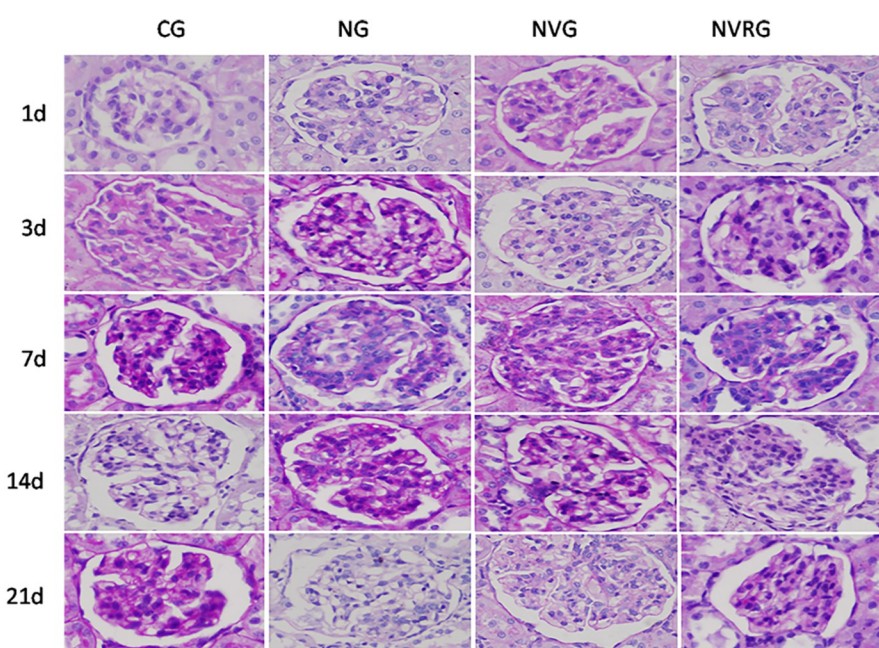

**Fig 3. PAS staining images of pathological changes in renal tissues among 4 groups (×400).**

performed immunohistochemistry to assess PCNA and mTOR expression. PCNA expression in mesangial cells appeared as brown-yellow granules located in the nucleus and was occasionally observed in the glomerular mesangial cells of normal rats. A summary of PCNA expression quantification is provided in Table 3. Compared with the CG, the NG group showed significantly increased PCNA expression in renal tissues (p <0.05) and in glomerular mesangial cells on days 1, 3, 7, 14, and 21. PCNA expression in the NG group increased over time, peaking on day 7, which aligns with the trend in mesangial cell proliferation. Compared with the NG group, both the NVG and NVRG groups exhibited significantly lower PCNA expression in glomerular mesangial cells at all time points (p <0.05), with a reduction in PCNA expression observed on days 14 and 21 compared to day 7 (Fig 4) (p <0.05).

mTOR expression in mesangial cells was visualized as nest-like, cord-like, or diffusely distributed brown-yellow granules located in the nucleus. The expression levels of mTOR across different groups and time points are summarized in Table 4. In the NG group, mTOR expression was significantly higher in renal tissues and glomerular mesangial cells compared to the CG group (p <0.05). In the CG group, there were no significant changes in mTOR expression

**Table 2. Comparison of renal tissue pathological damage grades across different groups at all time points.**

|  | Day 1 | Day 3 | Day 7 | Day 14 | Day21 |
|---|---|---|---|---|---|
| CG | 0.00±0.000 | 0.00±0.000 | 0.00±0.000 | 0.00±0.000 | 0.00±0.000 |
| NG | 0.67±0.516 | 1.33±0.516* | 2.83±0.408* | 2.00±0.632 | 0.83±0.754 |
| NVG | 0.83±0.408 | 1.00±0.632* | 1.83±0.752# | 1.17±0.408# | 0.67±0.516 |
| NVRG | 0.67±0.516 | 0.83±0.753* | 1.83±0.408# | 1.00±0.632# | 0.67±0.516 |

Note:

*p<0.05, indicates significantly severe renal pathological damage in comparison to CG starting from Day 3.

#p<0.05, indicates significantly alleviated pathological renal damage in NVG and NVRG on Days 7 and 14 compared to NG.

**Table 3. Comparison of PCNA expression grades across different groups at various time points.**

|  | day 1 | day 3 | day 7 | day 14 | day 21 |
|---|---|---|---|---|---|
| CG | 0.00±0.00● | 0.00±0.00 | 0.00±0.00▲ | 0.00±0.00 | 0.00±0.00 |
| NG | 3.41±1.23● | 5.44±2.09 | 8.14±2.65▲ | 6.58±1.86 | 2.92±1.51 |
| NVG■ | 3.62±2.97 | 4.33±0.67 | 4.81±1.16★ | 4.42±1.56 | 2.47±1.17 |
| NVRG■ | 2.57±1.07 | 3.53±1.39 | 5.27±2.18✳ | 4.75±2.17 | 2.11±1.63 |

Note: indicates PCNA expression in renal tissues on Day 1 after antibody injection and

●$p < 0.01$; indicates the highest PCNA expression in renal tissues on Day 7 after antibody injection and

▲$p < 0.01$; indicates a lower PCNA expression of NVG than that of NG on Day 7 and

★$p < 0.05$;

✳ indicates a decreased PCNA expression of NVRG on Day 7 and $p < 0.05$; indicates no significant difference in PCNA expression between NVG and NVRG at all time points and

■$p > 0.05$.

across the different time points (all $p > 0.05$). However, in the NVG and NVRG groups, mTOR expression was significantly lower compared to the NG group at all time points ($p < 0.05$), with mTOR expression levels decreasing over time (Fig 5).

## The expression of PCNA and mTOR correlates with the severity of renal pathology in Thy-1 nephritis rats

The relationship between PCNA or mTOR expression and renal pathological damage was analyzed, revealing that 24-hour urinary protein levels, PCNA expression, and mTOR expression were positively correlated with each other ($p < 0.05$), and all were positively correlated with the pathological damage grading ($p < 0.05$), as shown in Table 5.

## Discussion

Abnormal proliferation of mesangial cells, key intrinsic components of the glomerulus, along with the release of inflammatory mediators and the pathological accumulation of extracellular

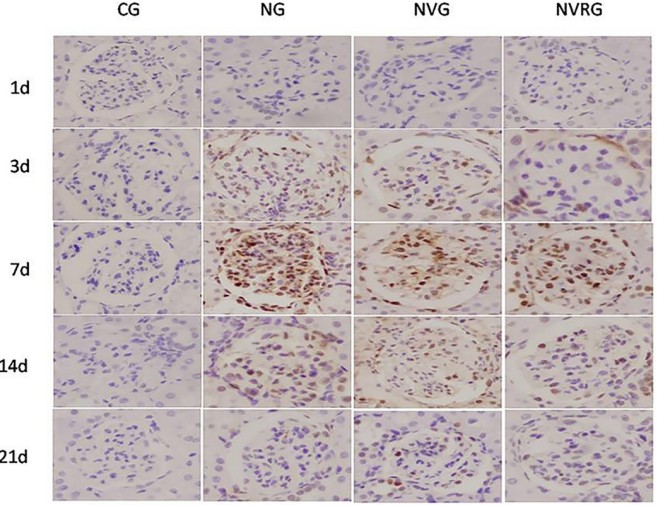

**Fig 4. Representative immunohistochemistry images of PCNA expression in renal tissues among CG, NG, NVG, and NVRG (×400).**

**Table 4. Comparison of mTOR expression grading among different groups at different time points.**

|  | day 1 | day 3 | day 7 | day 14 | day 21 |
|---|---|---|---|---|---|
| CG | 1.33±0.58● | 2.00±2.00 | 1.33±2.31▲ | 2.00±2.00 | 2.33±1.52 |
| NG | 6.00±3.00● | 8.00±1.00 | 12.00±0.00▲ | 8.00±1.00 | 3.33±1.15 |
| NVG■ | 4.67±1.15 | 4.67±1.15 | 10.00±1.00★ | 3.33±1.15 | 2.33±1.15 |
| NVRG■ | 3.81±1.15 | 3.91±2.31 | 10.00±3.00✻ | 4.46±1.15 | 3.00±2.00 |

Note: indicates mTOR expression in renal tissues on Day 1 after antibody injection and

●p<0.01; indicates the highest mTOR expression in renal tissues on Day 7 after antibody injection and

▲p<0.01; indicates a lower mTOR expression of NVG than that of NG on Day 7 and

★p<0.05;

✻ indicates a decreased mTOR expression of NVRG on Day 7 and p<0.05; indicates no significant difference in mTOR expression between NVG and NVRG at all time points and

■ p>0.05.

matrices, are critical factors leading to glomerular sclerosis and the progression of various forms of glomerulonephritis to end-stage renal disease [16, 17]. Therefore, developing new drugs and therapeutic strategies for glomerulonephritis that maintain a balance between mesangial cell proliferation, hypertrophy, and apoptosis while improving the metabolism of mesangial matrices is essential for delaying or reversing glomerular sclerosis. Previous studies have shown that $1.25(OH)_2D_3$ can inhibit mesangial cell proliferation and induce apoptosis by blocking the cell cycle [18, 19]; however, the underlying mechanism remains unclear. In this study, we established a nephritis rat model induced by Thy-1 injection and evaluated renal pathology to elucidate the roles and mechanisms of $1.25(OH)_2D_3$ in glomerulonephritis.

Compared with the CG, the experimental groups developed significant proteinuria on day 1 after antibody injection, which peaked on day 3 and returned to normal levels by day 14, likely due to the self-healing nature of the Thy-1 nephritis model. The NVG showed lower urinary protein levels than the NG at all time points, indicating that $1.25(OH)_2D_3$ could reduce proteinuria and protect the kidneys, consistent with the findings of Szeto CC [14]. Additionally, the NVG exhibited significantly milder histopathological changes compared to the NG at

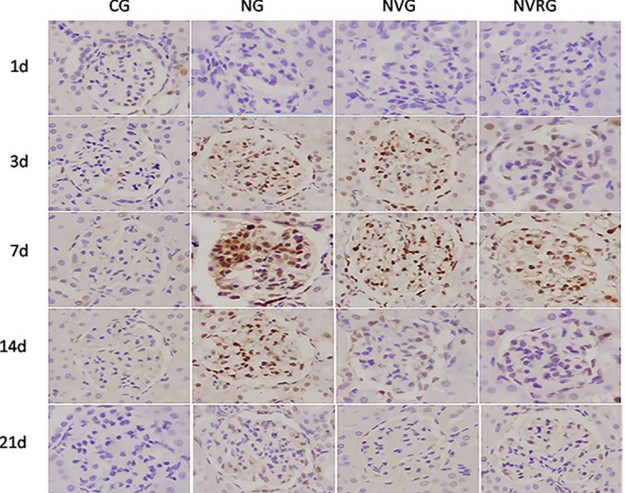

**Fig 5. Representative immunohistochemistry images of mTOR expression in renal tissues among CG, NG, NVG, NVRG (×400).**

**Table 5. Correlation among pathological damage grading, PCNA expression, and mTOR expression in glomerular renal tissues.**

|  | r | P |
|---|---|---|
| Pathological damage and mTOR | 0.729 | P<0.001 |
| Pathological damage and PCNA | 0.652 | P<0.001 |
| PCNA and mTOR | 0.677 | P<0.001 |

all time points, including reduced mesangial cell proliferation, decreased extracellular matrix accumulation, alleviated inflammatory factor buildup, and fewer balloon adhesions observed on day 7. These findings suggest that 1.25(OH)$_2$D$_3$ mitigates Thy-1 nephritis not only by reducing proteinuria but also by inducing favorable histopathological changes.

mTOR plays a crucial role in cell survival, proliferation, and apoptosis [20]. It is often continuously activated in many tumor cells, indicating its close relationship with the proliferation, survival, and metastasis of these cells [21]. A recent study demonstrated that mTOR is upregulated in mice with lupus nephritis and is strongly associated with the pathological proliferation of mesangial cells [10]. While 1.25(OH)$_2$D$_3$has been shown to induce differentiation and inhibit cell proliferation in various tissues, it remains unclear whether mTOR mediates this biological effect and whether there is a direct correlation between them. In a 2006 study, Regulska et al. found that mTOR is involved in the protection of nerve cells by 1.25(OH)$_2$D$_3$and its analogs [22], as well as in the inhibition of staurosporine-mediated apoptosis of osteoblasts by 1.25(OH)$_2$D$_3$and its receptors [23]. Currently, there are few reports on the relationship between mTOR and mesangial cell proliferation. In this study, mTOR expression in the experimental groups increased from day 1, peaked on day 7, and then gradually declined. Both NVG and NVRG had lower mTOR expression levels compared to NG at all time points (p <0.05), but there was no statistically significant difference between NVG and NVRG (p >0.05). The expression of mTOR was positively correlated with the expression of PCNA, a cell cycle regulatory protein, suggesting that 1.25(OH)$_2$D$_3$may inhibit mTOR expression and, thus, the proliferation of mesangial cells. We also observed that the combined intervention of rapamycin and 1.25(OH)$_2$D$_3$did not produce a synergistic effect in rats with nephritis. There was no significant difference in mTOR expression between rats treated with both agents and those treated with 1.25(OH)$_2$D$_3$alone, suggesting that rapamycin and 1.25(OH)$_2$D$_3$may share the same target: the mTOR protein. The specific mechanism by which 1.25(OH)$_2$D$_3$inhibits mTOR expression should be further explored from a molecular biology perspective.

Our comprehensive analysis of the effects of 1.25(OH)$_2$D$_3$ on renal histopathology, mTOR protein, and PCNA in rat nephritis models showed that the grading of pathological renal damage, PCNA expression, and mTOR expression were positively correlated (p <0.05). These findings suggest that mTOR and PCNA may be involved in the inhibition of mesangial cell proliferation and the alleviation of renal injury by 1.25(OH)$_2$D$_3$.

The present study has some limitations, as we did not explore the underlying molecular mechanisms in greater depth. In our future research, we plan to conduct PCNA and mTOR knockdown experiments to further validate their roles in the protective effects of 1.25(OH)$_2$D$_3$ against glomerulonephritis. Additionally, only a single dose of 1.25(OH)$_2$D$_3$ was used in this study, so different doses will be investigated in future studies.

In conclusion, our results suggest that 1.25(OH)$_2$D$_3$ significantly inhibits mesangial cell proliferation in rats, potentially through the reduction of mTOR and PCNA expression. The mTOR protein may be a key target in the biological effects of 1.25(OH)$_2$D$_3$.

## Author Contributions

**Conceptualization:** Jian-feng Li, Lei Jin.

**Data curation:** Lei Jin.

**Formal analysis:** Lei Jin.

**Funding acquisition:** Jie Suo.

**Validation:** Rui Yang.

**Visualization:** Xiao-ping Yang.

**Writing – original draft:** Huan Ma.

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
