## [Decision Letter · Decision Letter 0]

15 Jul 2024

PONE-D-24-226681.25(OH)2D3 decrease PCNA and mTOR expression and alleviates renal injury in Thy-1 nephritis rat modelPLOS ONE

Dear Dr. Yang,

Thank you for submitting your manuscript to PLOS ONE. After careful consideration, we feel that it has merit but does not fully meet PLOS ONE’s publication criteria as it currently stands. Therefore, we invite you to submit a revised version of the manuscript that addresses the points raised during the review process.

We look forward to receiving your revised manuscript.

Kind regards,

Santhi Silambanan, MD, DNB

Academic Editor

PLOS ONE

A clean copy of the edited manuscript (uploaded as the new *manuscript* file)”.

4. PLOS requires an ORCID iD for the corresponding author in Editorial Manager on papers submitted after December 6th, 2016. Please ensure that you have an ORCID iD and that it is validated in Editorial Manager. To do this, go to ‘Update my Information’ (in the upper left-hand corner of the main menu), and click on the Fetch/Validate link next to the ORCID field. This will take you to the ORCID site and allow you to create a new iD or authenticate a pre-existing iD in Editorial Manager. Please see the following video for instructions on linking an ORCID iD to your Editorial Manager account: https://www.youtube.com/watch?v=_xcclfuvtxQ".

6. Thank you for stating the following financial disclosure: 

 [This work was supported by the National Natural Resources Foundation of China (81160090) and Research Project Supported by Xinjiang Production and Construction Corps of China (2009GG59)].  

7. In the online submission form, you indicated that [Due to space constraints, the original data cannot be fully reflected in the text. Due to format limitations (the original data is in Excel format), it cannot be uploaded to the submission system as a supplementary file. If necessary, please contact the corresponding author to provide.]. 

8. Your ethics statement should only appear in the Methods section of your manuscript. If your ethics statement is written in any section besides the Methods, please move it to the Methods section and delete it from any other section. Please ensure that your ethics statement is included in your manuscript, as the ethics statement entered into the online submission form will not be published alongside your manuscript. 

9. We note that Figure(s) 2, 3, 4 and 5 in your submission contain copyrighted images. All PLOS content is published under the Creative Commons Attribution License (CC BY 4.0), which means that the manuscript, images, and Supporting Information files will be freely available online, and any third party is permitted to access, download, copy, distribute, and use these materials in any way, even commercially, with proper attribution. For more information, see our copyright guidelines: http://journals.plos.org/plosone/s/licenses-and-copyright.

a. You may seek permission from the original copyright holder of Figure(s) 2, 3, 4 and 5  to publish the content specifically under the CC BY 4.0 license. 

10. Please include your tables as part of your main manuscript and remove the individual files. Please note that supplementary tables (should remain/ be uploaded) as separate ""supporting information"" files.

Reviewers' comments:

Reviewer's Responses to Questions

**Comments to the Author**

1. Is the manuscript technically sound, and do the data support the conclusions?

Reviewer #1: Yes

Reviewer #2: Yes

2. Has the statistical analysis been performed appropriately and rigorously? 

Reviewer #1: Yes

Reviewer #2: Yes

3. Have the authors made all data underlying the findings in their manuscript fully available?

Reviewer #1: Yes

Reviewer #2: Yes

4. Is the manuscript presented in an intelligible fashion and written in standard English?

Reviewer #1: Yes

Reviewer #2: No

5. Review Comments to the Author

Reviewer #1: Thank you for the opportunity to review your excellent and important paper. I have just a few minor comments.

1. In the Abstract you mention "thylakoid cells" in the Objective. Should this have been "glomerular mesangial cells"?

2. I would have loved to see Tables III and IV in graphs, but understand if there wasn't enough room.

3. In Table V, you don't need the lower left triangular region, which is simply a reflection of the upper right triangular region.

4. Do you care to speculate on the relative efficacy of Vitamin D vs rapamycin? Has rapamycin already been shown to be effective by itself? For example, does this paper show that? Lock, Helen R., Steven H. Sacks, and Michael G. Robson. "Rapamycin at subimmunosuppressive levels inhibits mesangial cell proliferation and extracellular matrix production." American Journal of Physiology-Renal Physiology 292.1 (2007): F76-F81.

Reviewer #2: This is an interesting piece of work. The findings provide more insight on the role of 1.25(OH)2D3 on glomerulonephritis. The authors should check that the article is clear and concise. Grammatical errors were present in the abstract and manuscript.

6. PLOS authors have the option to publish the peer review history of their article (what does this mean?). If published, this will include your full peer review and any attached files.

Reviewer #1: No

Reviewer #2: No

---

## [Author Response · Author response to Decision Letter 0]

28 Aug 2024

Thank you for carefully reviewing our manuscript.

Weare grateful to all reviewers for their constructive critique.

Thank you for your recognition and encouragement.

---

## [Editor Report · Decision Letter 1]

1 Sep 2024

PONE-D-24-22668R11.25(OH)2D3 decrease PCNA and mTOR expression and alleviates renal injury in Thy-1 nephritis rat modelPLOS ONE

Dear Dr. Yang,

Thank you for submitting your manuscript to PLOS ONE. After careful consideration, we feel that it has merit but does not fully meet PLOS ONE’s publication criteria as it currently stands. Therefore, we invite you to submit a revised version of the manuscript that addresses the points raised during the review process.

**ACADEMIC EDITOR: **

The responses to the reviewers have been found to be adequate.

The article needs to undergo few spell checks and reframing of sentences, for better presentation of the study. 

Kind regards,

Santhi Silambanan, MD, DNB

Academic Editor

PLOS ONE

Journal Requirements:

Additional Editor Comments:

I appreciate the authors for addressing the reviewers' queries. But the manuscript needs to have spell and grammar checks. These need to be addressed.

---

## [Author Response · Author response to Decision Letter 1]

3 Sep 2024

Thank you very much for the speed and attitude of the reviewers and editors, and for your positive comments on our manuscript.

Being able to publish our research in PLOS ONE is one of the most important things for our research team this year. Therefore, we attach great importance to each point raised by reviewers and editors, we try to respond the editor's comments as perfect and clear as possible. We have also done a lot to this work.

---

## [Editor Report · Decision Letter 2]

10 Sep 2024

1.25(OH)2D3 decrease PCNA and mTOR expression and alleviates renal injury in Thy-1 nephritis rat model

PONE-D-24-22668R2

Dear Dr. Xiao-ping Yang,

We’re pleased to inform you that your manuscript has been judged scientifically suitable for publication and will be formally accepted for publication once it meets all outstanding technical requirements.

Kind regards,

Santhi Silambanan, MD, DNB

Academic Editor

PLOS ONE

Additional Editor Comments (optional):

The authors have adequately addressed the queries raised by the reviewers and the editor.
---

## [Editor Report · Acceptance letter]

17 Sep 2024

PONE-D-24-22668R2 

PLOS ONE

Dear Dr. Yang, 

I'm pleased to inform you that your manuscript has been deemed suitable for publication in PLOS ONE. Congratulations! Your manuscript is now being handed over to our production team.

Kind regards, 

on behalf of

Dr. Santhi Silambanan 

Academic Editor

PLOS ONE